# Observing Snow Cover and Water Resource Changes in the High Mountain Asia Region in Comparison with Global Mountain Trends over 2000–2018

Claudia Notarnicola 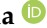

Eurac Research, Institute for Earth Observation, 39100 Bolzano, Italy; claudia.notarnicola@eurac.edu

**Abstract:** The quantification of snow cover changes and of the related water resources in mountain areas has a key role for understanding the impact on several sectors such as ecosystem services, tourism and energy production. By using NASA-Moderate Resolution Imaging Spectroradiometer (MODIS) images from 2000 to 2018, this study analyzes changes in snow cover in the High Mountain Asia region and compares them with global mountain areas. Globally, snow cover extent and duration are declining with significant trends in around 78% of mountain areas, and the High Mountain Asia region follows similar trends in around 86% of the areas. As an example, Shaluli Shan area in China shows significant negative trends for both snow cover extent and duration, with −11.4% (confidence interval: −17.7%, −5.5%) and −47.3 days (confidence interval: −70.4 days, −24.4 days) at elevations >5500 m a.s.l. respectively. In spring, an earlier snowmelt of −13.5 days (confidence interval: −24.3 days, −2.0 days) in 4000–5500 m a.s.l. is detected. On the other side, Tien Shan area shows an earlier snow onset of −28.8 days (confidence interval: −44.3 days, −8.2 days) between 2500 and 4000 m a.s.l., governed by decreasing temperature and increasing snowfall. In the current analysis, the Tibetan Plateau shows no significant changes. Regarding water resources, by using Gravity Recovery and Climate Experiment (GRACE) data it was found that around 50% of areas in the High Mountain Asia region and 30% at global level are suffering from significant negative temporal trends of total water storage (including groundwater, soil moisture, surface water, snow, and ice) in the period 2002–2015. In the High Mountain Asia region, this negative trend involves around 54% of the areas during spring period, while at a global level this percentage lies between 25% and 30% for all seasons. Positive trends for water storage are detected in a maximum 10% of the areas in High Mountain Asia region and in around 20% of the areas at global level. Overall snow mass changes determine a significant contribution to the total water storage changes up to 30% of the areas in winter and spring time over 2002–2015.

**Keywords:** snow cover; water resources; mountain; High Mountain Asia; climate change

---

## 1. Introduction

Mountains cover 22% of the earth surface, they are home to around 13% of global population, and provide between 60% and 80% of the global fresh water including drinking, domestic use, irrigation, industry and hydropower [1,2]. Under climate change conditions, mountain areas are considered sentinels as most phenomena can be amplified in this environment [3,4]. In particular, the changes related to snow and glacier mass for both the Northern and Southern Hemispheres are a clear sign of the climate change impact in mountain areas [5–7]. The importance of mountains is recognized by several initiatives such as the one of the United Nations that included mountain ecosystem conservation in its Sustainable Development Goals, and by GEO-GNOME with the aim to identify Essential Climate Variables (ECVs) for high altitudes and to define essential mountain variables [8].

Several studies monitored and quantified snow changes worldwide. Recently, Notarnicola, (2020) [9] analyzed the snow parameter trends in the last two decades by using NASA-Moderate Resolution Imaging Spectroradiometer (MODIS) products and reported that, at a global level, around 78% of the areas with significant changes are suffering from snow cover and duration negative trends. Considering different elevation belts, at a medium elevation between 1000 and 4000 m a.s.l., positive and negative changes are intermixed, while at an elevation higher than 4000 m a.s.l., only negative changes are found in the different snow parameters. Areas located in western USA, South America and Australia are suffering from extensive decline in many parameters, while areas in Northeastern Russia, Northern Europe and in some parts of central Asia also show positive trends [5,10–12].

This analysis of 18 years of MODIS data is consistent with longer time series such as in Hori et al. (2017) [13]: they found that, based on 38 years of data, western Eurasia has a strong snow cover duration (SCD) decrease up to two months over this time period; some areas of Eastern Asia and North America have a weaker decline, and in some cases they show an SCD increase of 1 month over three decades. Moreover, the snow cover changes show a seasonal dependency, as observed by Zhang and Ma, (2018) [14] over Eurasia for the period 1972–2006, where the significant decreases are detected during spring and summer and are less strong in autumn and winter. Therefore, the snow melt occurs earlier while snow onset does not change very significantly in this period [9].

Bormann et al. (2018) [15] compared long-term analysis of Snow Cover Extent from National Oceanic and Atmospheric Administration-(NOAA-SCE) data with fractional snow cover extent from the algorithm known as MODSCAG (MODIS Snow-Covered Area and Grain size) in four mountain regions. They identified areas with retreating snow cover extent in western Himalaya, Australia and western USA, as well as areas with slight snow extent increase, like Karakorum, that showed a positive mass balance for glaciers as well [16].

All these studies show a general decline of snow resources in mountain areas, whereas at regional level different trends may exist. Among the areas with contrasting changes, it is important to address the High Mountain Asia (HMA) region as it plays a relevant role, with its basin being one of the most populated in the world. This region, which encompasses the HMA, is also named the Third Pole because it hosts the largest frozen water resources outside the polar regions. As it is widely addressed in several studies, this area's environmental conditions have changed in the last century [17,18].

Across 10 major catchments in HMA, based on data from 1987 to 2009, trends in snow water equivalent are found not uniform with the decreasing trend in the period from March to August, while some areas in the Pamir–Tien Shan region have a significant increase in the period from December to February [6]. This can be related to changes in the Winter Westerly Disturbance, which resulted in increased snow fall in this part of the Indus. Moreover, the snow behavior is related to the altitude, with midelevation areas of HMA showing stronger negative trends than higher or lower elevation. At lower elevation, some positive snow water equivalent trends are also found in the December–February period. The overall snow volume decline is confirmed on a longer period from 1987 to 2016 [19]. In this region, a time series of combined active and passive remotely sensed data shows a trend of earlier snow melt with few exceptions in the Karakorum Mountains and western Kunlun Mountains. There is a strong dependency on temperature with a rate of 4.5 days/degree [20].

Tien Shan Mountains reveal similar spatial heterogeneity when using MODIS products: SCD values mainly decrease in the central and eastern Tien Shan by 11.88% and 8.03%, respectively, while the northern and western Tien Shan mountains have an increase of 9.36% and 7.47%, respectively. The major factor influencing these snow cover changes is the temperature trend [21]. The changes in snow cover and glaciered areas reflect on the changes in runoff regime, where total water storage experiences a decline in the middle and east part and a slight increase in the western part with a total average rate of around −3.7 mm/y [22].

In the Tibetan Plateau, Wang et al. (2017) [23] identified no clear trends for snow parameters such as SCD, snow onset and melt, by using MODIS products in the period from 2000 to 2015. The part with elevation below 3500 m a.s.l. experiences a shorter SCD (with later snow onset and earlier

melt), while regions in the central and southwestern part of the Tibetan Plateau exhibit longer SCD (with earlier onset and delayed melt). Temperature and precipitation have both a role to explain these changes, and their impact is stronger at higher elevations, even though for a short period of time, no clear trend is detected by Huang et al. (2016) [24] and Huang et al. (2017) [25], but a strong interannual variability is found [26]. Considering a longer period from 1980 to 2018, instead, signals of snow decline were detected [27]. By exploiting long time series of ground measurements, a slight increase in snow cover is found from 1951 to 1997 and then a slight decrease from 1997 to 2012 [28–30]. In the period from 1976 to 2006, changes in seasonality are also found indicating a significant decreasing trend for snow cover in the western HMA in summer and autumn, and for the southern HMA in all seasons. On the other side, a significant increasing trend of snow cover is detected in the central and eastern HMA in autumn, winter, and spring [26,31,32]. In the Tibetan Plateau snow cover changes are strongly related to water resources such as lake level changes. In the period from 2001 to 2010, by using MODIS data, Zhang et al. (2012) [33] detected a significant relationship between lake water level changes and the related basin snow cover area (SCA) in four main lake basins. It is worthwhile mentioning that in this area, as well in other mountain areas, cloud cover can hamper the detection of snow cover, so that a dedicated processing of images is required to improve the accuracy of the detected trends [9,34].

The 2013 IPCC report (IPCC 2013) [35] and the European Environmental Agency report (EEA 2016) [36] mentioned the snow cover changes at a global–continental scale and the related impact on water availability for downstream areas as key elements to be monitored. In fact, changes in the snow and glacier accumulation may have cascade effects from mountain to lowlands, impacting sectors such as drinking water supplies, agriculture, irrigation, hydropower and tourism. For mountains, the term "water tower" is used to describe their role of water storage and supply for environmental and human demands in different sectors. Immerzel et al. (2020) [37] developed a Water Tower Index with the aim to rank mountain areas based on their water-supplying role and on their vulnerability related to both environmental, social and economic threats. The results indicated that water towers in Southern America and Asia are more vulnerable than areas in Northern America and Europe. In particular, the Indus basin is found as the most important and vulnerable water tower.

The impact of climate change on water availability in snow-dominated regions can affect more than one-sixth of the Earth population for their water supply. On global scale the largest changes in the hydrological cycle due to warming are expected in the snow-dominated basins and especially at mid-higher elevation [38]. Analyses conducted at a global level by using Gravity Recovery and Climate Experiment (GRACE) data across 13 years (2002–2016) indicate a high variability in total water storage (TWS), influenced by a mix of factors such as human activities, interannual variability and climate changes. In the Tibetan Plateau, the changes can be mainly ascribed to water level increases in relation to increased precipitation rates and glacier melting [39,40]. In general, the HMA area is very relevant for water resources as the relative drainage basins are among the world's most dynamic, complex, and intensively populated areas, responding to demands of around 1.3 billion people. Considering the increase of the population, water supply is under extreme threat, due to natural processes and variations related to climate changes [41].

In this context, a quantification of snow changes and water availability is a key element to understand the current processes and take mitigation actions for future scenarios.

While a recent study by Pullianinen et al. (2020) [42] addressed the snow mass changes at a continental scale, this work focuses on the snow cover and related water resources in global mountain areas.

The main aim of this study is to provide a quantification of snow cover changes in the HMA areas, the link to the main meteorological drivers, and the impact on water resources. All the results in the HMA region are presented through a comparison with trends for snow and water resources in mountain areas at a global level. Information is obtained by exploiting the full time series of MODIS products at the highest ground resolution of 500 m, and by using several snow parameters such as

Snow Cover Area (SCA), Snow Cover Duration (SCD), First Snow Day (FSD), Last Snow Day (LSD) and Snow Line Altitude (SLA), and their analyses in different seasons and elevation belts. With respect to Notarnicola (2020) [9], this work provides a focus on the HMA areas with full details on the snow cover changes. As this area is one of the most vulnerable water towers, it is relevant to monitor the changes in the snow cover and phenology for different seasons and elevation belts. Moreover, the total water resources are evaluated by using GRACE data from 2002 to 2013, and for the first time the contribution of snow changes to TWS is quantified specifically in mountain ranges at global level. The main purpose is to monitor the area/region in which the detected snow cover changes strongly affect water resources.

## 2. Materials and Methods

### 2.1. Study Area

This study focused on mountain areas in the HMA, that stretch from the Pamir and the Hindu Kush in the west to the Mishmi Hills and Shaluli Shan mountains in the southeast, from Kunlun and Qilian mountains in the north to the Himalayas in the south. With an average elevation of around 4000 m a.s.l., it is the largest and highest mountain area in the world hosting all the peaks above 7000 m a.s.l. The extension of the HMA area that was considered in this study is the following: 58–122° E and 9–43° N as stated in www.icimod.org/?page=43 [41]. In the area definition, the Tien Shan areas were also included and analyzed in the following because they share the same Water Tower Unit with Pamir, Karakorum and the Tibetan Plateau [37].

The mountain area extensions identified in the HMA areas were extracted from the reference layer Global Mountain Biodiversity Assessment (GMBA) [43]. This inventory reports the name for each of the 1003 identified areas, which is very relevant for the comparison among their characteristics. The selection of the polygons to be investigated was conducted by considering yearly SCA averaged over 18 years, larger than 10%, as well as SCD averaged over 18 years, longer than 20 days. This threshold was also considered for different elevation belts and seasons and was mainly applied to remove from statistics the impact of regions with ephemeral snow. As a result of this selection, the number of mountain areas in the HMA regions was 45. In this study the snow parameter and water resource trends in the HMA areas were compared with the behavior at a global level. In the latter case, the number of total areas was 383, of which 351 were located in the Northern Hemisphere (these areas include the ones in the HMA) and 32 in the Southern Hemisphere. The extent of the study area in the HMA region and at a global level is illustrated in Figure 1. The distribution of areas based on mean latitude and elevation for the global and HMA mountain regions is reported in Figure 1c.

### 2.2. Data

#### 2.2.1. MODIS Products

The main snow related parameters analyzed in this study were: SCA, SCD, FSD, LSD, SLA. These parameters were calculated from the MODIS Daily L3 Global 500 m Grid, Version 6 (MOD10A1.006) products, available at https://nsidc.org/data/mod10a1. The whole period from 2000 to 2018 was exploited considering a hydrological year interval set as follows: 1st October–30th September for the Northern Hemisphere and 1st April–31st March for the Southern Hemisphere. Together with yearly information, SCA data were analyzed for the main seasons: September–October–November (SON), December–January–February (DJF), March–April–May (MAM), June–July–August (JJA) and in different elevation belts.

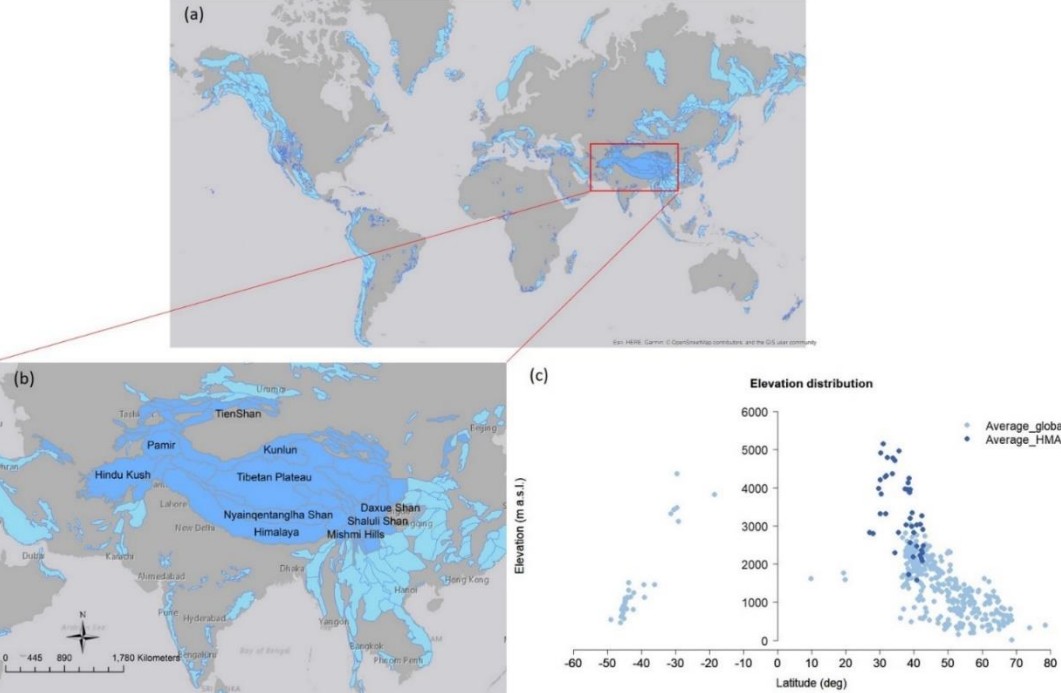

**Figure 1.** Extension of the study areas analyzed in this paper: the global mountain area extent (**a**) and the High Mountain Asia (HMA) mountain extent (in darker blue) (**b**). The extent for the global and HMA mountain areas is derived from the Global Mountain Biodiversity Assessment (GMBA) reference layer. From this layer, only areas with average snow cover area (SCA) > 10% and average snow cover duration (SCD) > 20 days were analyzed, thus excluding areas with ephemeral snow presence. For the HMA region, some of the main mountain ranges are reported in the figure. (**c**) Distributions of the global and HMA areas for average elevation and latitude. The latitude and elevation values are the averages calculated over the extension of the single area of interest.

### 2.2.2. Meteorological Parameters

Two main meteorological parameters, near surface air temperature (hereafter called temperature) and snowfall, were analyzed as the main explanatory variables for the snow dynamics. Temperature and snowfall data were derived from MERRA-2 dataset available from 1980 to 2018 at $0.625° \times 0.5°$ longitude by latitude grid resolution [44]. The products from the MERRA2 model were already used in several studies showing good performances [7,9,45]. Some studies assessed the performances of MERRA2 data specifically over the HMA region. Considering temperature and snowfall in relation to snow cover, MERRA2 data seem to have a strong performance with respect to other reanalysis data because they assimilate snow depth data over the Tibetan Plateau. In this area, it is particularly relevant to assimilate ground data to consider the transient snowpack [46]. On the other side, Liu and Margulis (2019) [47] pointed out that MERRA2 data, as well as other reanalysis products, show an underestimation of snowfall precipitation over HMA areas. It is worthwhile mentioning that the complex orography and the lack of in-situ data at high elevations remain as the two main limitations in the accuracy assessment of these data. Panahi and Behrangi (2019) [48] reported very good performances of MERRA2 for snow accumulation in the comparison with snow water equivalent information.

With respect to MODIS products, the main limitation of these datasets was related to the coarse ground resolution of MERRA2 which may hinder the detection of phenomena at small and local scale. Keeping in mind this limitation, the focus of the analysis was not on the spatial signal, but rather on the relative changes in time and relationship to snow cover changes [49,50]. For this reason, the analysis with meteorological parameters was not carried out at different elevation belts.

### 2.2.3. GRACE Data for Total Water Storage (TWS)

Another important parameter in mountain areas related to snow cover dynamics is water availability. To monitor changes in water availability, data from the GRACE satellite were exploited. The data derived from GRACE satellite mission provide information on the variations in the terrestrial TWS based on measured temporal variations of the Earth's gravity field [51]. The estimates of TWS from GRACE include the variations of groundwater, soil moisture, surface water, snow and ice, and have been already successfully used in hydrological studies, including estimation of river discharge relationships [52], and for understanding the impact of glacier and ice cap changes [53]. Moreover, as snow plays a relevant role for TWS changes, several studies focused on the derivation of snow water equivalent by disentangling snow contribution from other parameters such as ground water, soil moisture, evapotranspiration [54–58]. In the HMA area, GRACE data have been extensively used to understand the increase of water mass in relation to different components such as water storage, lake volume, snow water equivalent, glacier mass, soil moisture [39,59].

GRACE Tellus Monthly Mass Grids used in this study represent monthly gravitational anomalies with respect to a 2004–2010 time-mean baseline [60]. The GRACE dataset is produced with three different formulations produced by CSR (U. Texas/Center for Space Research), GFZ (GeoForschungsZentrum Potsdam), and JPL (NASA Jet Propulsion Laboratory). Each center is part of the GRACE Ground System and generates the Level-2 data (spherical harmonic fields) used in this dataset. In this study the CSR dataset was used, since it was already exploited in relation to snow cover changes, providing good results in comparison with GLDAS model outputs [61]. Due to the sampling and postprocessing of GRACE observations, surface mass variations at small spatial scales tend to be attenuated. Therefore, the data were multiplied by a scaling grid available at NASA/GRACE/MASS_GRIDS/LAND_AUX_2014.

GRACE measurements are affected by errors which depend on the own measurement system, model errors in the processing system to remove atmospheric and ocean signals, and spatial leakage error associated with a limited range of spherical harmonics. Some authors provide estimates of these errors at monthly level in the Artic region, indicating that they may be below 5 mm for GRACE measurements, 10 to 50 mm for atmosphere and ocean model errors, and about 10 mm for leakage errors [62]. Addressing specifically mountain areas, Chen et al. (2016) [22] provided an estimated error in the entire Tien Shan mountain areas of around 30 mm, while Yang and Chen (2015) [61] derived an uncertainty of around 0.12 cm/y. Moreover, the comparison with ground data to assess the accuracy of GRACE data is quite challenging because of the unavailability of reference field measurements in each study region and for each component of TWS. Notwithstanding these limitations, several studies have used the GRACE dataset and demonstrated the reliability for monitoring TWS changes in different areas worldwide [22,63–65].

### 2.2.4. Auxiliary Data

Several auxiliary data were exploited in the analysis presented in the paper:

- Digital elevation model: The Global Multi-resolution Terrain Elevation Data 2010 (GMTED2010, version Breakline Emphasis) dataset was exploited [66]. The ground resolution is 7.5 arc-seconds resolution, while the vertical root mean square error is estimated to be between 26 and 30 m.
- Percentage of forest cover: this was derived from the MODIS product MOD44B (MODIS Vegetation Continuous Fields—Percentage of tree cover) [67]. The updated map of the year 2015 was used in this analysis.
- Water bodies mask: the water bodies classification derived from MODIS Land Cover Type product (MCD12Q1) [68].

As the main aim of the study was to compare the snow variability between HMA and global mountain areas, the auxiliary layers were selected because globally available and consistent with the MODIS products used to derive snow parameters.

## 2.3. Methods

The main snow parameters: SCA, SCD, FSD, LSD and SLA were derived from MODIS products MOD10A1.006 using the approach presented in Notarnicola (2020) [9]. In the following, a summary of the derived parameters is reported. For full details, the readers can refer to Notarnicola (2020) [9]. All the processing and statistical analyses were performed using the online platform Google Earth Engine, where different MODIS products and GRACE data are available at global scale for the whole period of availability. The platform also allows an easy ingestion of external datasets, for both the raster and shape file format [69].

SCA values represent fractional snow cover and were derived from NDSI_Snow_Cover layer present in MOD10A1.006 by applying the formulation of Salomonson and Appel (2016) [70]. SCD values report the number of days a pixel is covered by snow in a hydrological year, while FSD and LSD values indicate the first and the last date in the hydrological year that a pixel is snow covered, respectively. For the derivation of these three parameters, a second-order autoregressive approach was used to reduce the impact of cloud cover which may hinder the proper calculation [9,71].

The approach was applied for each hydrological year and for the Northern and Southern Hemisphere separately. To check the validity of the approach and determine the final accuracy of the derived parameters, SCD, FSD, LSD, a comparison with ground measurements and model simulation were carried out. Considering that global characteristics for snow cover can be highly variable, a detailed sensitivity analysis was performed to select the most appropriate thresholds for both ground snow depth and fractional snow cover values [72]. The selected threshold for fractional snow cover is 50% while for the ground snow depth the threshold is adaptive and depends on the maximum snow depth in the area [9]. The comparison with ground measurements reported the following values for correlation coefficient (R), Mean Absolute Error (MAE) and bias:

SCD: R = 0.84 ($p < 0.01$) with MAE = 21.1 days, and a bias of $-3.1$ days ($n = 466$).
FSD: R = 0.93 ($p < 0.01$) with MAE = 11.1 days, and a bias of 4.7 days ($n = 466$).
LSD: R = 0.89 ($p < 0.01$) with MAE = 13.9 days, and a bias of $-2.2$ days ($n = 466$).

The value $n$ indicates the number of time series for the analyzed ground measurements.

A slight worsening of the performances is mainly detected for the high percentage of missing information (>70%) due to both cloud cover and polar darkness. In this case of a large amount of missing values, especially for the challenging case of the polar darkness, a comparison was done with simulations derived from the model Global Land Data Assimilation System (GLDAS) to further test the validity of the interpolation approach and to overcome the limitation of ground measurements that can be representative of very local scale [73]. The results are very consistent with the statistical values obtained at global scale in different conditions of missing values, as reported above [9].

For the analysis with yearly and seasonal SCA values, no interpolation procedure for cloud reduction was considered, because in case of averages of a large number of images (around 90 images for the three-monthly averages and on 365 for the yearly averages), the impact of cloud coverage is strongly reduced. To check the validity of such assumption, the yearly and 3-monthly SCA mean values were compared with the results of the averaged data with MOD10A2 products, which provide the 8-day maximum snow extension [11,34]. In this case, the SCA differences ranged from 5% to 15%, and on a yearly base the difference was on average around 6%, thus indicating to be within the known accuracy of the fractional snow cover products [70]. From yearly SCA values, the SLA values were derived. The SLA represents the average altitude in an area where snow is detected for a specific year and it was calculated as the 20th percentile over the altitude distribution of the yearly SCA values for each mountain polygon [7,74,75].

The above-mentioned parameters were statistically analyzed to understand the behavior over the period of availability from 2000 to 2018. To characterize the snow dynamics, the analyses were carried out both for single pixel value and for area averages in different elevation belts. Moreover, together with yearly average, the averages for four main seasons, SON, DJF, MAM, JJA were assessed.

For the statistical analysis, several methods were adopted. First, for evaluating the presence of a monotonic increasing or decreasing trend in the analyzed variables (snow parameters for the period 2000–2018 and TWS values for the period 2002–2015), the nonparametric Mann–Kendall (M–K) test was adopted, as this approach is not so sensitive to the presence of outliers. The M–K approach used in the study is the form used for monotonic trend without seasonal components, and the significance of the trends was obtained by using a Z value at α level of significance. For the linear regression pixel and area based, the slope (change per unit time) is computed through the nonparametric procedure proposed by Sen (1968) [76]. From the slope, the total change for a specific parameter during the observed period was obtained by multiplying the slope by the number of years [77]. A 100(1 − α)% two-sided confidence interval on the slope estimate is obtained by the nonparametric technique using a normal distribution. In the text, each parameter change was reported along with 95% confidence interval in brackets.

The dependency of SCA on temperature and snowfall was investigated by using a Pearson correlation between the dependent parameter (SCA during SON, DJF, MAM, DJF) and the explanatory variables (temperature and snowfall). As a second approach, the general dominance weights approach proposed by Budescu (1993) [78] was used to quantify the importance of one explaining variable (temperature or snowfall) with respect to the other. In this approach, a coefficient of prediction $R^2$ was calculated for a multiple linear regression by using temperature and snowfall as predictors. This $R^2$ value from the multiple regression was then compared with the $R^2$ of the single regressions. Through this comparison, the weight of the different explanatory variable was assessed. The significance of this regression was evaluated with a two-tailed F-test, setting the significance threshold at 5% level.

## 3. Results

### 3.1. Snow Dynamics in the HMA Areas over 2000–2018

Yearly changes and rate of changes for the main snow parameters over the HMA region are illustrated in Figure 2. The results indicate quite heterogenous areas with positive and negative changes intermixed. Significant negative changes are mainly found for the areas of Shaluli Shan (for the names of the areas please refer to Figures 1, 5 and 14), southeast of the Tibetan Plateau.

Tibetan Plateau does not have a clear trend—indeed, only 3% of pixels have a significant trend at 5% level [77]. The areas with the highest number of significant pixels at 5% level are Mishmi Hills and Nyainqentanglha Shan with 17% and 13% significant negative trend pixels for SCA changes and 15% and 13% for SCD changes. Moreover, for these areas, the highest percentage of significant negative trend pixels is located in the elevation belt of 4000–5500 m a.s.l. The percentage of pixels with significant positive trends is always below 10% for all the areas.

Starting from the analysis per pixel shown in Figure 2, the snow dynamics were investigated considering area averages to understand which areas are most affected by changes, at which elevation belt and season. For sake of comparison, the results for the HMA region were then compared with those derived from global mountain ranges [9]. For the analysis across elevation belts, the comparison started with 1000 m a.s.l. because only two areas have part of their territory in the elevation belt of 300–1000 m a.s.l. The comparison was done for all snow parameters SCA, SCD, FSD, LSD, SLA for elevation belts from 1000 m a.s.l. upward (Figure 3).

The results from HMA areas and at global level show very similar trends with a predominance of negative ones. While at a global level some positive trends are still visible at lower-medium elevations up to 4000 m a.s.l., in the HMA regions, for SCA, SCD and LSD, there are only negative trends. Some positive trends are found for FSD, that indicate a delay in the season start regardless. It is interesting to mention that for areas above 5500 m a.s.l., 67% of the global areas are located in the HMA region, and as it was observed from global analysis the trends are negative for all parameters, with a stronger impact on SCA and SCD.

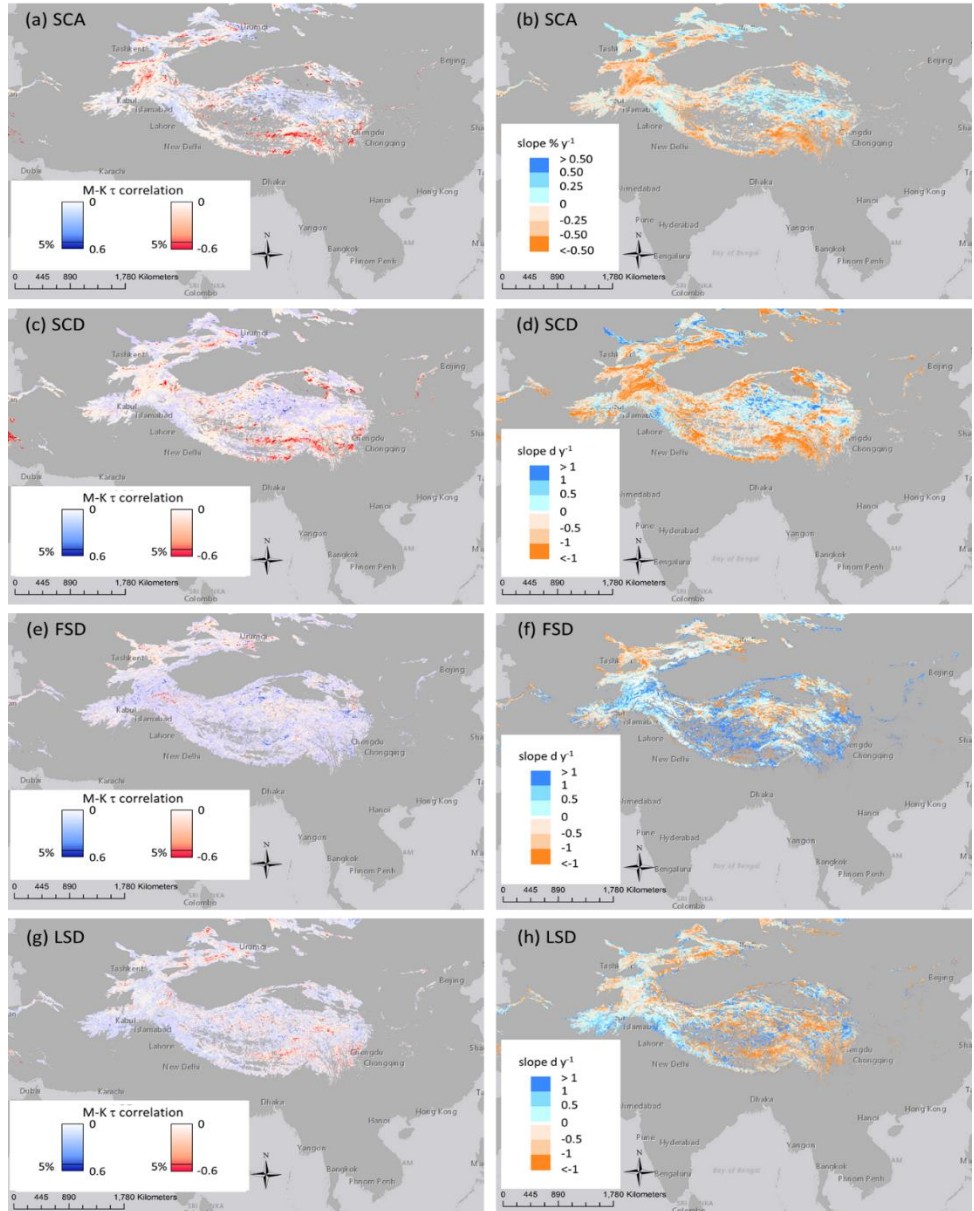

**Figure 2.** Yearly changes (Mann–Kendall (M–K) τ correlation of the monotonic trend of the variable for the period 2000–2018) and rate of changes (Sen's slope of the monotonic trend of the variable for the period 2000–2018) for the main snow parameters: (**a,b**) SCA, (**c,d**) SCD, (**e,f**) first snow day (FSD), (**g,h**) last snow day (LSD) using the NASA-Moderate Resolution Imaging Spectroradiometer (MODIS) products from 2000 to 2018 over the HMA area. These images were extracted from the global ones after Notarnicola (2020). In the images of yearly changes, 5% (dark blue) and 5% (dark red) indicate those pixels with increasing or decreasing trends in time at 5% significance level based on M–K τ correlation statistics. For FSD, positive changes and positive slopes indicate a delay in the start of the snow season. All the statistics at a pixel level are calculated for those pixels with 18 points (a value per year).

The area of Khrebet Talasskiy Alatau shows all parameters with positive trends in the elevation belt from 300 to 1000 m a.s.l. This area is located in the western part of the Tien Shan mountain which shows an increase in SCA and SCD when compared with the eastern and southern part [21,22].

A similar situation is found for the analysis of SCA changes over the different seasons (Figure 4). Negative trends dominate throughout the seasons, with peak of 18.5% of the areas for elevation higher than 5500 m a.s.l. in DJF period. At the same elevation, some positive trends at 10% significance level can be identified during DJF and MAM period. A general overview of the snow changes over HMA

region is presented in Figure 5, where the different colors represent the number of parameters in the above mentioned analysis with at least a significant change (percentages of significant trends at pixel level, yearly mean change, seasonal change, elevation belt change). In the region, 29 areas showed significant changes of which 86% with negative or mainly negative behavior.

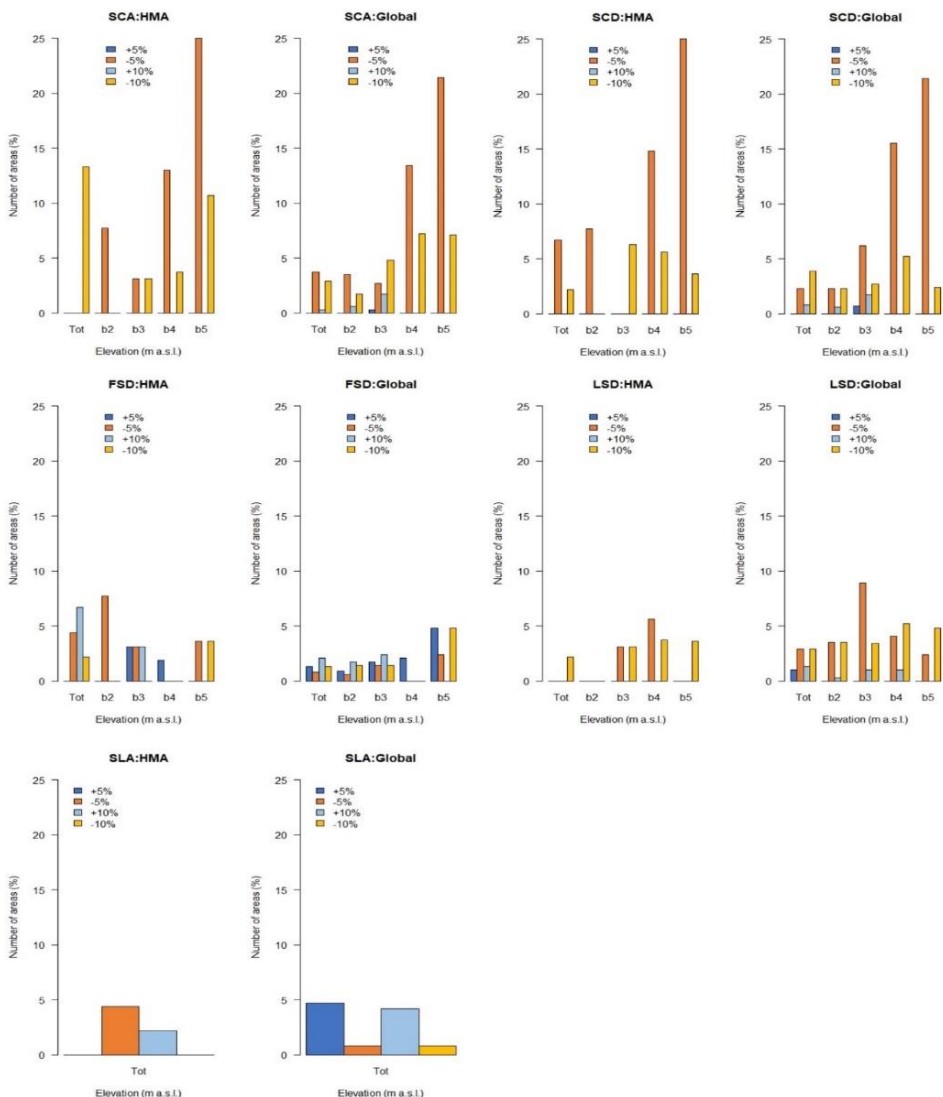

**Figure 3.** Number of areas (%) with respect to the total number of areas with a significant positive and negative change at 5% (+5%/−5%) and 10% (+10%/−10%) level for the main snow parameters SCA, SCD, FSD, LSD, snow line altitude (SLA) analyzed in this study. For FSD a positive/negative change represents a delay/advance in the snow onset. For SLA, a positive/negative change represents an increase/decrease in the snow line altitude. Tot represents the whole elevation range, b2: 1000–2500 m a.s.l., b3: 2500–4000 m a.s.l., b4: 4000–5500 m a.s.l., b5: >5500 m a.s.l.

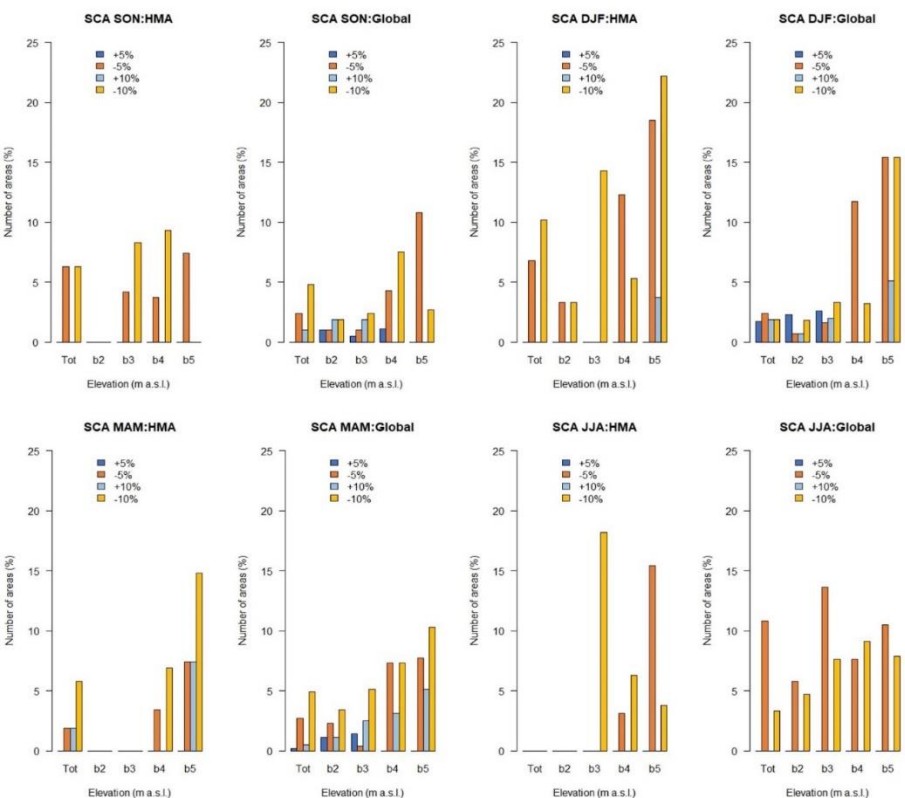

**Figure 4.** Number of areas (%) with respect to the total number of areas with a significant positive and negative change at 5% (+5%/−5%) and 10% (+10%/−10%) level for the different seasons: September–October–November (SON), December–January–February (DJF), March–April–May (MAM), June–July–August (JJA). Tot represents the whole elevation range, b2: 1000–2500 m a.s.l., b3: 2500–4000 m a.s.l., b4: 4000–5500 m a.s.l., b5: >5500m a.s.l.

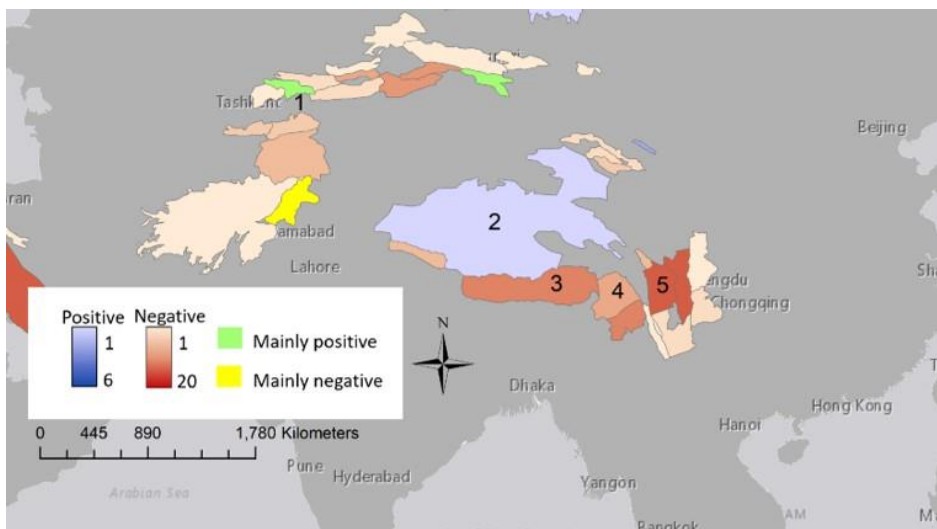

**Figure 5.** For each area, the figure reports the number of parameters with at least a significant change (percentages of significant trends at pixel level, yearly mean change, seasonal change, elevation belt change). The color bars in the legend represent the number of parameters. The numbers in the figure refer to specific areas mentioned in the text (1) Khrebet Talasskiy Alatau; (2) Tibetan Plateau; (3) Nyainqentanglha Shan; (4) Mishmi Hills; (5) Shaluli Shan.

### 3.2. Relationship to Meteorological Variables

In this section, the relationship between SCA changes and meteorological parameters was investigated in the main seasons to understand which are the main drivers for snow dynamics in the last 18 years. In a first step, SCA values in the different seasons were correlated with temperature and snowfall separately, to monitor how changes in SCA can be related to changes in temperature and snowfall. For all the areas in HMA, SCA changes are negatively correlated with temperature, i.e., an increase (decrease) of SCA is correlated with a decrease (increase) of temperature, while SCA changes is positively correlated with SF. Temperature changes dominate in all the seasons, especially in DJF and JJA, while during SON there is a higher percentage of areas correlated with snowfall changes (Figure 6). This is further confirmed from the predominance analysis (Figure 7) where for HMA areas, temperature dominates in all seasons except in the SON season, when around 40% of the areas are governed by snowfall changes [6,21]. With respect to global areas, from SON to JJA, the importance of temperature increases and reaches the peak in JJA period. This can be related to the melting time during MAM and JJA periods. In fact, in the HMA region there is a high number of areas at elevations >4000 m a.s.l. where the melting period is later in the season, on average in June and July.

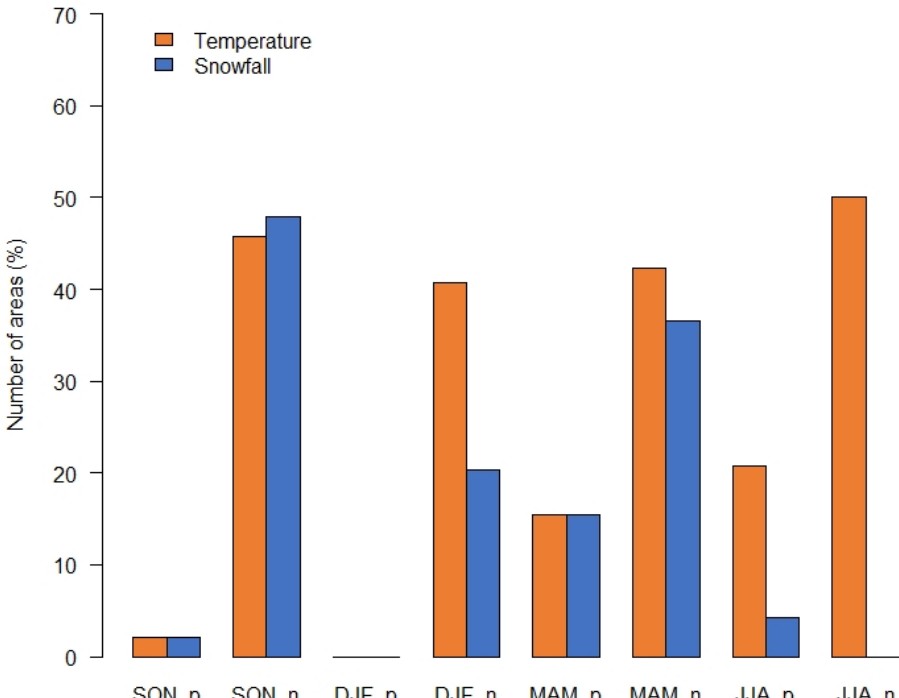

**Figure 6.** Number of areas (%) with respect to the total number of areas in the HMA region with a significant relationship at 5% level between SCA and temperature/snowfall changes for the different seasons SON, DJF, MAM, JJA. SON_p, SON_n indicates an increasing/decreasing trend respectively for SCA in the SON period. The same symbology applies for the other seasons.

### 3.3. Impact on Water Availability

One of the main important parameters in mountain areas is water availability, and to investigate this parameter GRACE data in terms of TWS values were analyzed in the period 2002–2015 for both HMA and global areas. This period was selected because of the data availability. Then, the changes in TWS were correlated to SCA changes at yearly base and throughout the different seasons.

The resulting changes and rate of changes for TWS are illustrated in Figures 8 and 9 for global areas and HMA areas, respectively. At a global level, there are several areas with significant positive and negative changes consistent with the trends indicated in Rodell et al. (2018) [40]. Positive hotspots are identified in northern Europe, north-eastern Russia, central USA and the Tibetan Plateau, while negative

hotspots can be found in the Gulf of Alaska Coast, in the Middle East, in the areas of Greater Caucasus and Zagros Mountains, in the Himalaya areas and in South America. The values of TWS changes can vary greatly among the areas. For example, Saint Elias Mountains and Chugach Mountains in the Gulf of Alaska coast show a TWS decrease of −161.7 cm (−181.2 cm, −141.1 cm) and of −167.2 cm (−189.8 cm, −139.7 cm), respectively. Among the positive areas, Zapadno-Skhalisnkiy Khrebet in northeastern Russia on the Oktotsk sea has an increase of 29.3 cm (22.4 cm, 33.1 cm).

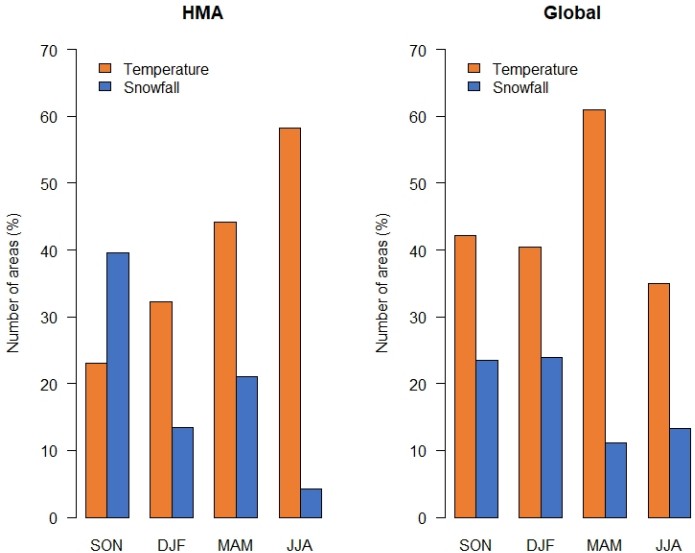

**Figure 7.** Results of the predominance analysis as SCA dependent from temperature and snowfall for the HMA and global region. Only the significant results at 5% level are reported.

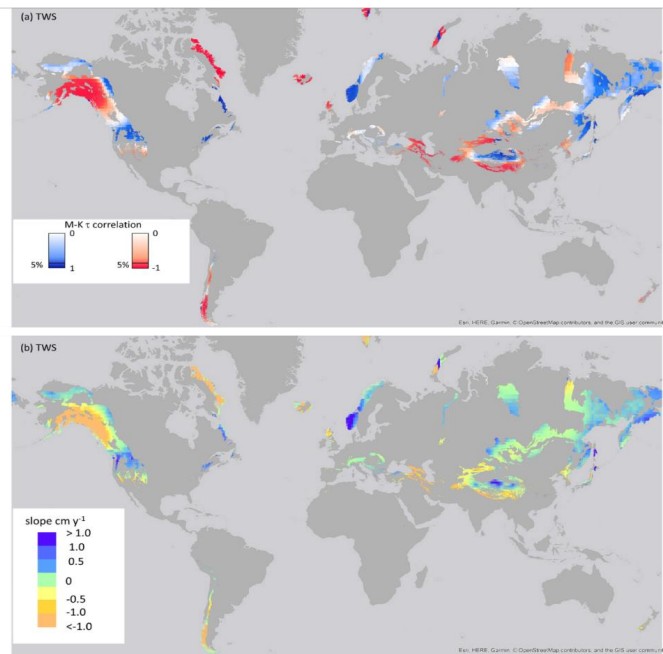

**Figure 8.** Global view for (**a**) the yearly total water storage (TWS) changes (M–K τ correlation of the monotonic trend of TWS for the period 2002–2015) and (**b**) rate of changes (Sen's slope of the monotonic trend of TWS for the period 2002–2015) by using GRACE data.

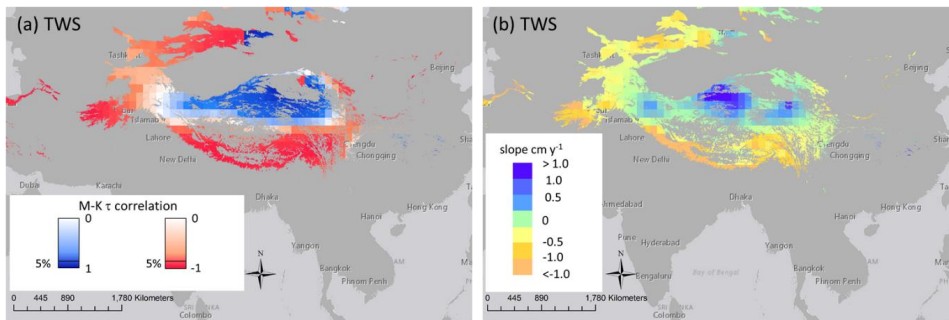

**Figure 9.** (**a**) TWS yearly changes (M–K τ correlation of the monotonic trend of TWS for the period 2002–2015) and (**b**) rate of changes (Sen's slope of the monotonic trend of TWS for the period 2002–2015) from GRACE data for the HMA areas.

In the HMA region (Figure 9), the area of the Tibetan Plateau shows a significant increase of water while the Tien Shan areas (in the northern part) and the Himalaya region show (in the southern part) significant negative trends. As examples, the central part of Tien Shan has a decrease of −8.2 cm (−12.9 cm, −6.3 cm) while the Tibetan Plateau has a positive trend with an increase of 2.6 cm (−0.2 cm, 6.5 cm). All the above-mentioned examples are significant at 5% level, except the Tibetan Plateau that is significant at 10% level. The results obtained from GRACE about TWS are the sum of different components which may change constantly, like the increase/decrease of ice field and/or snow melt, the rate of precipitation and the interannual variability with transition from wet to dry period (or vice versa) due to climatic cycles (such as El Niño, La Niña) over the 14 year period of the analysis [40].

TWS changes were investigated also considering area averages. The areas with positive and negative changes in the period 2002–2015 are reported in Figure 10. Around 50% of areas in the HMA region and 30% at global level are suffering from significant TWS decrease. In HMA region, this decrease involves around 54% of the areas during MAM period, while at a global level the percentage of areas remains between 25% and 30% for all the seasons. Positive trends are found for maximum 10% of the areas in HMA region and for more than 20% of the areas at global level.

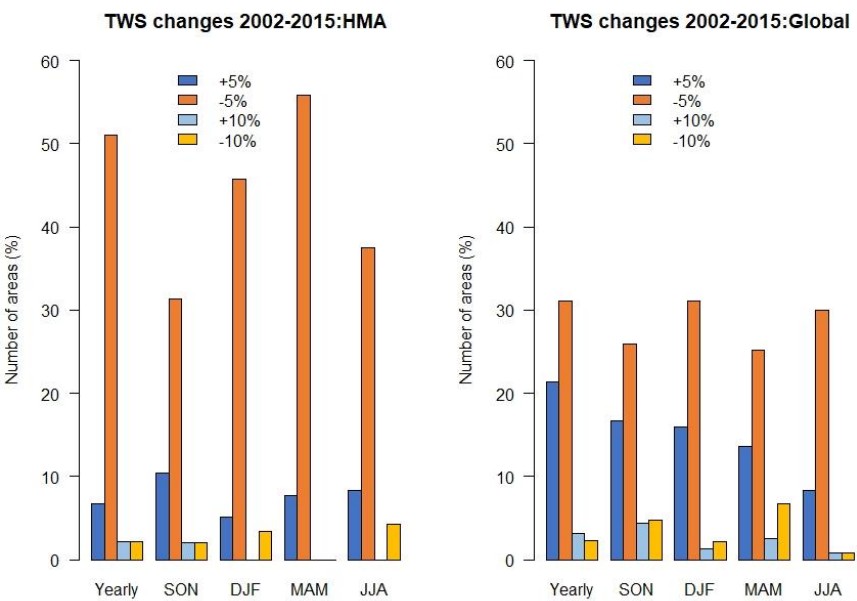

**Figure 10.** Number of areas (%) with respect to the total number of areas with yearly and seasonal significant changes (at 5% and 10% significant level) for TWS over the HMA and global mountain areas. A significant positive and negative change at 5% level is represented with +5%/−5%, respectively and 10% level with +10%/−10%, respectively.

Considering the rate of changes (slope) for areas significant at 5% level, the negative values can reach up to −2.5 cm/y in the JJA period at global level (Figure 11). For HMA region, the lowest value is found during JJA period with a value of −0.9 cm/y. Highest positive rates are found mainly during MAM and JJA periods which represent the melting period for most of the areas [40].

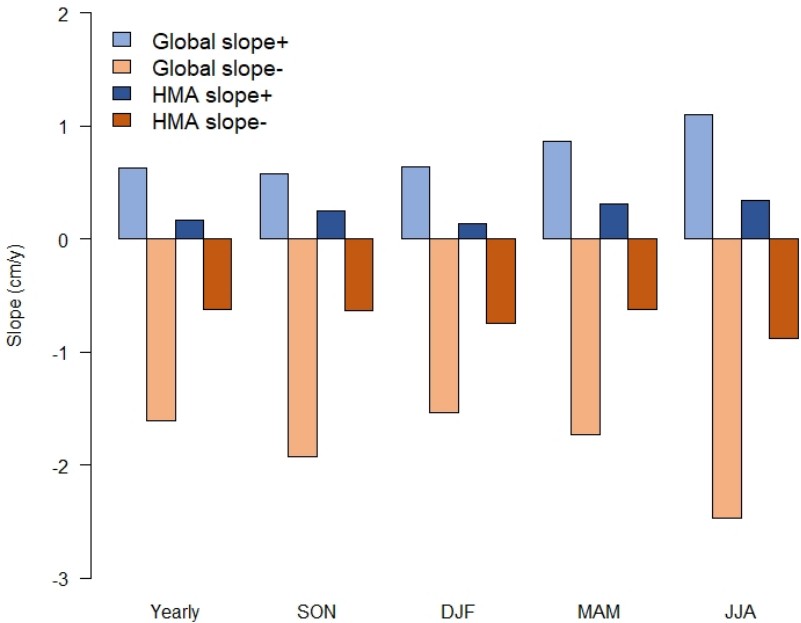

**Figure 11.** Average TWS rate of changes (Sen's slope of the monotonic trend of TWS for the period 2002–2015) for HMA and global areas, at yearly base and throughout the different seasons.

As a next step, the main objective of this study is to understand in which areas the TWS fluctuations can be strongly dependent on the snow mass changes. One of the main contributors to TWS variability in mountain areas is related to snow cover dynamics for both intra-annual and interannual time period [56,79]. A general relationship between TWS and SCA values was investigated to understand the variations during seasons (SON, DJF, MAM, JJA). Considering that the TWS values are the sum of several components such as snow mass (expressed for examples as snow water equivalent), soil moisture, ground water, the influence of SCA (and then of snow water equivalent change) has to be dependent on the different seasons as shown in Figure 12. Here the average behavior of TWS and SCA values during seasons is illustrated.

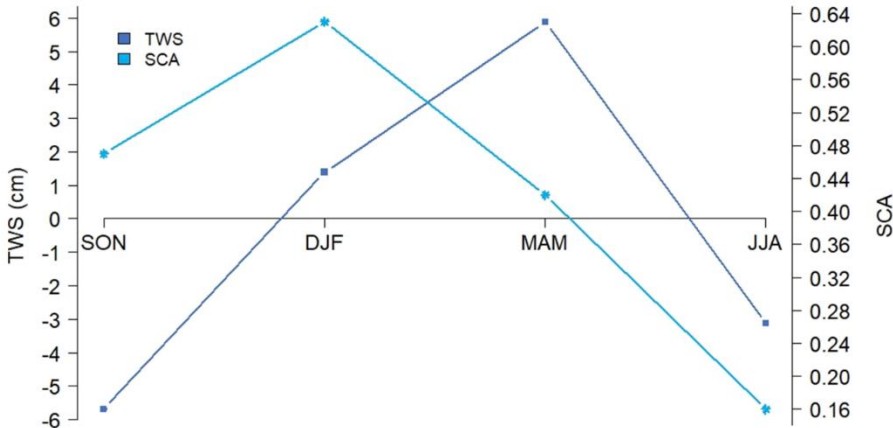

**Figure 12.** Average behavior between TWS and SCA values during different seasons.

As observed by Niu et al. (2007) [56] and Wang et al. (2017) [58], seasonal variations of TWS are greatly influenced by the snow mass showing positive anomalies in winter and spring due to snow accumulation and melt, and negative anomalies in summer. Seasonal variations of below-ground water storage are opposite to those of snow mass and the TWS, showing negative anomalies in winter due to groundwater drainage and positive anomalies in spring due to infiltration of snowmelt water.

The observed changes in TWS were correlated to SCA changes for the yearly and seasonal mean to understand when a correlation exists between these two parameters, thus expecting the impact of snow cover changes on water availability.

Several areas have a positive significant correlation of TWS with SCA changes, indicating that increase/decrease in TWS is correlated to increase/decrease of SCA (Figure 13). For areas in the Northern Hemisphere and the HMA, the highest number of areas are found during the melting period in MAM and JJA period. During MAM in HMA areas, around 22% of the areas show a significant correlation with SCA. In Southern Hemisphere, the highest number of areas is found (around 30%) in DJF which corresponds to summer, that is as well a melting period in these areas. Of course, several factors may contribute to TWS values as they count for ground water, soil moisture, wetland, and precipitation. The areas where the correlation between TWS and SCA is found significant at the 5% level are illustrated in Figure 14, where also the fact that one area can have significant correlation in more than one season is considered. It is worthwhile mentioning that areas with a yearly positive/negative correlation between TWS and SCA show the same trend (positive/negative) of correlation during the different seasons, apart from one area, Omineca Mountains (central USA), which shows a negative correlation in winter (DJF) and a positive one in spring (MAM). Areas in western USA are characterized by a dependence in the seasons DJF and MAM, while in Alaska and in northeastern Russia the dependency is stronger during start of the season in the SON period. SON and DJF are as well the main seasons for correlation between SCA and TWS for South America, where they correspond to the spring–summer period. Among mountain ranges in arid regions, Zagros Mountains show significant correlation between SCA and TWS from DJF to JJA period. In HMA region, the southern part (Daxue Shan, Shaluli Shan, Nyainqentanglha Mountains) has a dependency in DJF and MAM, while in the northern part (Narat Shan, Borohoro Shan in the Tien Shan area) there is a dependency in winter (DJF) and in summer period (JJA).

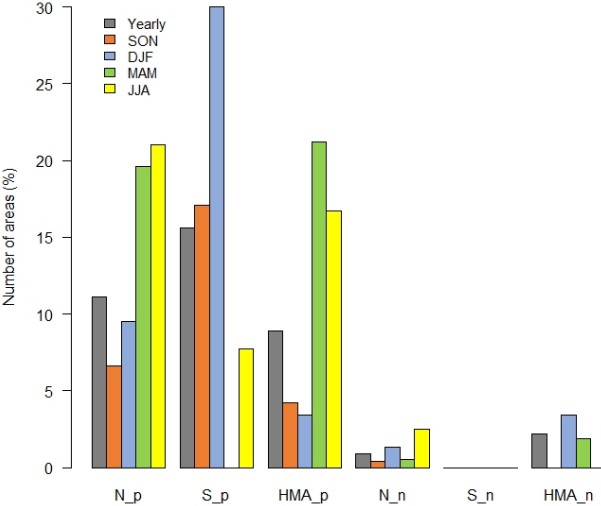

**Figure 13.** Number of areas (%) with respect to the total number of areas which show a significant correlation at 5% level between TWS and SCA changes for the main seasons. To better highlight this behavior in different regions, the areas have been divided among Northern (N), Southern Hemisphere (S) and HMA region (HMA). N_p, S_p, HMA_p indicate areas with a positive correlation between TWS and SCA, N_n, S_n, HMA_n indicate areas with a negative correlation between TWS and SCA.

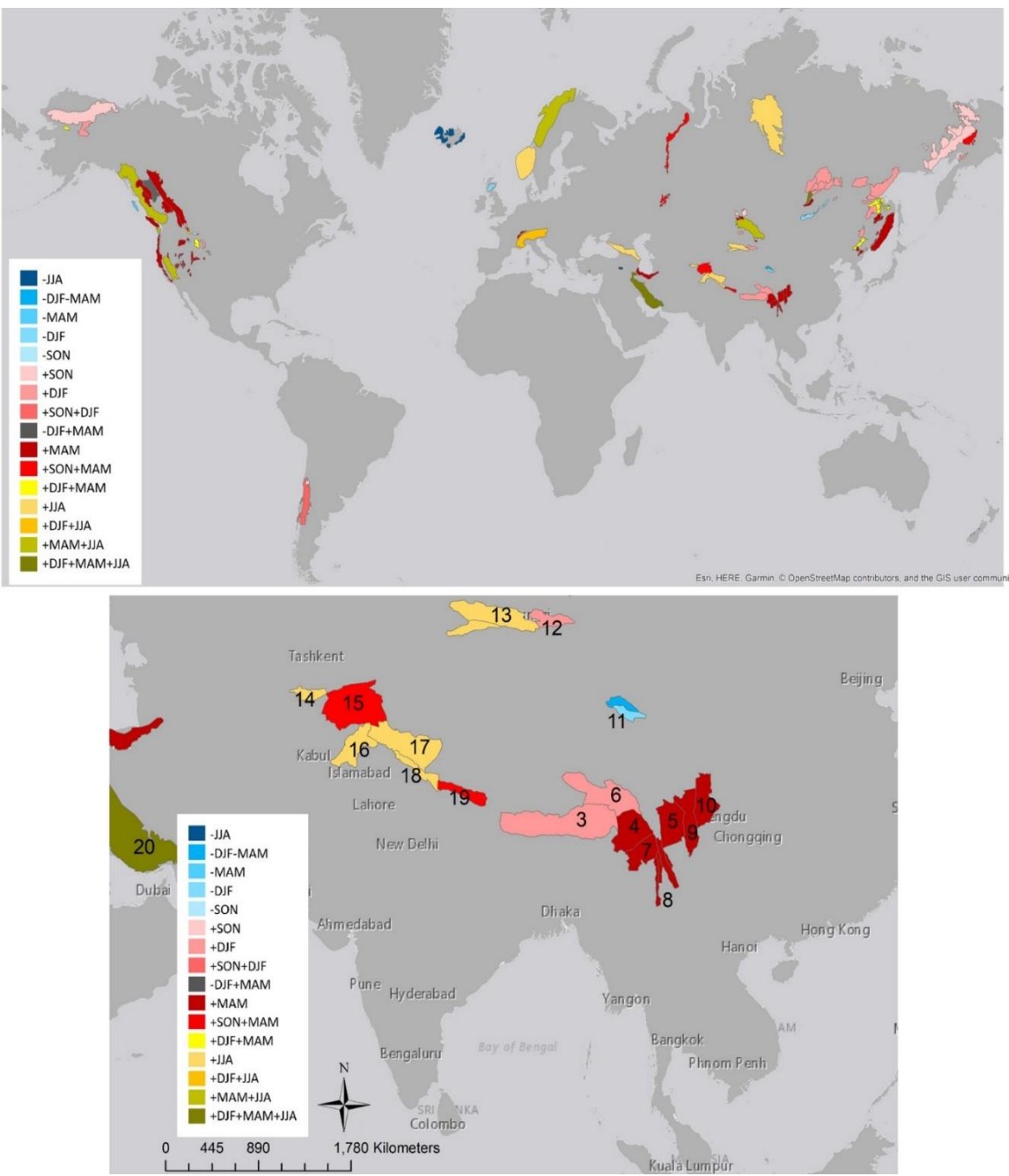

**Figure 14.** Areas with significant correlation at 5% level between TWS and SCA changes in different seasons at global level and in the HMA region. In the legend the sign '−' before the name of a season indicates a negative correlation between SCA and TWS in that specific season. The same applies for the sign '+'. In the HMA region: (3) Nyainqentanglha Shan, (4) Mishmi Hills (5) Shaluli Shan, (6) Tanggula Shan, (7) Patkai Hills, (8) Gaoligong Shan, Nu Shan, Hengduan Shan, Xue Shan, (9) Daxue Shan, (10) Qionglai Shan, (11) Qaidam Shan, Tergun Daba Shan, (14) Ghissarskiy Khrebet, (15) Pamir Mountains, (16) Malakand range, (17) Karakorum, (18) Ladakh range, (19) Nganglong Kangri. Outside the HMA region at the border of the defined rectangle 58–122° E and 9–43° N: (12) Bogda Shan, (13) Borohoro Shan-Narat Shan. As a comparison also (20) Zagros Mountains are indicated. (The areas are numbered in continuity with Figure 5).

## 4. Discussion

The presented analyses highlight a high variability of snow cover in the HMA region, where three main subregions can be identified. In the southern part, mainly negative trends are dominating, especially above 4000 m a.s.l. elevation. Daxue Shan (China) and Shaluli Shan (China) areas indicate a snow decline for both SCA and SCD at elevation >5500 m a.s.l., with −7.2% (−9.5%, −4.5%) and −27.2 days (−37.4 days, −15.4 days), −11.4% (−17.7%, −5.5%) and −47.3 days (−70.4 days, −24.4 days), respectively. These two areas show an earlier snow melt at an elevation of 4000–5500 m a.s.l., with LSD changes of −11.0 days (−24.3 days, −0.2 days) and −13.5 days (−24.3 days, −1.98 days). Nyainqentanglha Mountains (China) as well have SCA and SCD changes in the 18 years of −5.0% (−10.5%, −0.8%) and −18.5 days (−38.5 days, −3.8 days), respectively [23]. These three main areas are governed by temperature changes in winter (DJF) and spring (MAM) and by snowfall for the snow onset (SON). In the HMA central part with main reference to the Tibetan Plateau area, contrasting changes are shown where only a significant slight decrease of the snow line altitude is found. All the other snow parameters do not show significant changes [23]. In the northern part of the HMA region, positive and negative changes are intermixed in different parts of the area and, also for different elevation belts. Tien Shan area shows an earlier snow onset of −28.8 days (−44.3 days, −8.2 days) between 2500 and 4000 m a.s.l., governed by decreasing temperature and increasing snowfall. On the other side, at elevations between 4000 m a.s.l. and 5500 m a.s.l., an earlier snow melt is detected with −26.2 days (−41.6 days, −9.1 days). Another interesting area located in the western part of Tien Shan Mountain is Khrebet Talasskiy Alatau. This area presents contrasting changes on different elevation belts. At lower elevation up to 1000 m a.s.l., it is characterized by increasing SCA of 6.7% (2.2%, 11.1%), with a delay in snow melt of 17.9 days (2.3 days, 33.6 days). When moving at higher elevation, between 4000 m a.s.l. and 5500 m a.s.l., a SCA decrease of −15.2% (−23.3%, −2.2%) is detected, mainly occurring during wintertime (DJF). Increasing trends in the HMA areas, especially delayed snow melt, are also found in the Karakorum Mountains and western Kunlun Mountains over a longer time series 1978–2018 [80]. The HMA region is influenced by several climatic systems such as the Indian Summer Monsoon and the Winter Westerlies Disturbances which may result in contrasting trends [6,21,77]. As an example, the Tien Shan is characterized by increasing temperature with faster rate in the middle and east Tien Shan with a decrease in precipitation, while on the western part the temperature increase was not so strong in the last decades, showing an increase of 23% in precipitation in the last 55 years. This behavior aligns with the proposed increase strength of the Winter Westerlies Disturbances in the last 30 years over this side of the region [6]. Another important factor to be considered is the shift in precipitation from snow to rain which was stronger in the middle and east Tien Shan area with a reduction around 25% of snow fraction. Tien Shan mountains are very relevant as they are considered the water tower of Central Asia, providing fresh water for agricultural and other economic sectors [22,81]. Heavy pollution and aerosol contamination are also found to cause changes in the snow dynamics. In fact, the impact on albedo can enhance certain processes such as snowmelt [7,12,38]. These factors together with water vapor and radiative flux change are supposed to be the reason of the Elevation-Dependent-Warming effect [3] which indicates an increasing temperature with elevation.

As mountains are the water reservoirs for the downstream areas, the analyses of the water resources can provide information on the status of the availability. The results based on GRACE TWS data indicate a quite heterogenous behavior both at global scale and in the HMA region. As these data are related to the TWS, several factors need to be considered for finding a possible explanation of the changes. In some areas, the analyses reveal strict correlation with changes in snow and glacier mass changes such as in the northeastern coast between Canada and Alaska, where the water loss can be consistent with the fact that Canada subarctic lakes can suffer from snow decrease in this area [40]. The dependence on decreasing ice field is confirmed in South America, together with a multiyear drought from 2002 to 2010 which may have impacted the trends from 2002 to 2015.

In the HMA region, the increase of water level in the Tibetan Plateau can be related to the fact that a large part is endorheic and disconnected from downstream demand. This trend is confirmed by other satellite data such as Landsat, and this may be due to increase of precipitation and melting from ice fields. The rapid melting of glaciers driven by temperature increase already from the 1990s can contribute to the decrease water level in the Tien Shan area, while in the Himalaya areas the decreasing glacier mass cannot fully explain the changes. In this case, another important factor is related to water depletion for irrigation purposes [40,81].

The cross correlation of TWS and SCA changes has the objective to identify in which areas the snow contribution is correlated with TWS changes. Examples of yearly SCA and TWS variability for some areas in the HMA region are shown in Figure 15. Snow contributes in most of the areas to changes in TWS with a higher dependency on DJF and MAM period, when an increase of water quantity is expected due to snow accumulation and melting. In some areas, such as northeastern Russia, this dependency is anticipated to SON because the snow accumulation at this latitude may start earlier (Figure 14). In some cases, as the Tien Shan area, there is a negative correlation between TWS and SCA at yearly base, while when moving to single season the correlation remains positive in some subregions (Narat Shan and Borohoro Shan) (Figures 14 and 15). In this case, some authors identified a decreasing trend for precipitation in the overall period of analysis and an increasing temperature especially in the period from 2007, when the negative TWS values started [40,61,82]. Many authors attributed changes in TWS in Central Asia to agricultural water consumption, groundwater withdrawal and artificial reservoir [83,84]. The fact that in 20% of the areas TWS values seem to strongly depend on snow variability jointly with snow decline in the last two decades puts the availability of water resources at risk. As mentioned above, for the analysis in different areas, the relation between SCA and TWS changes needs to be carefully addressed, as several components of the water cycle can contribute to the GRACE signal. GRACE data have been already used to derive information on snow mass such as snow water equivalent in regions dominated by snow (e.g., areas at higher latitude and Arctic regions), if values of ground water are estimated through land surface models [56]. Frappart et al. (2006, 2011) [54,55] compared the annual cycles of GRACE-derived snow water equivalent retrievals with those derived from other sources such as satellite microwave observations, global land surface models, climatologies of snow depth, snowfall and snow water equivalent. Their results over North America illustrate a relatively good agreement between GRACE snow retrievals and existing snow mass products. On the other side, also other water components can be relevant as found by Trautman et al. (2018) [85], showing that at a northern latitude, not only snow but also liquid water (soil moisture and retained water) can be drivers for the TWS changes. Baharami et al. (2019) [79], showed that Canadian basins are strongly dominated by snow cover during the season. In 13 over 15 of the analyzed basins, snow water equivalent represents at least 20% of the TWS with a maximum of 30% when all the basins are combined.

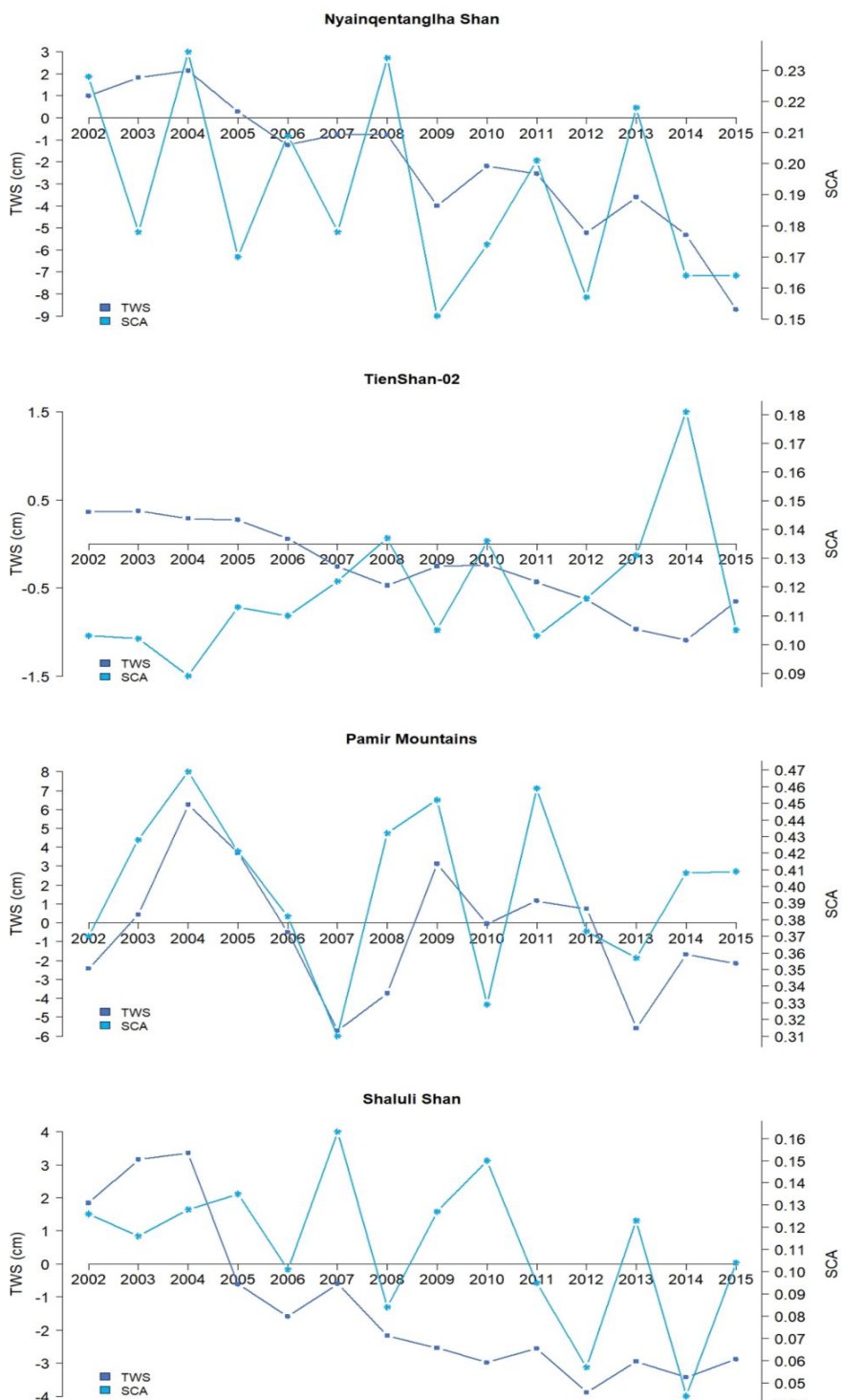

**Figure 15.** Time series of TWS and SCA values on a yearly base for some areas in the HMA regions. All the areas show a positive correlation between TWS and SCA except Tien Shan-02 (Western part of Tien Shan area), where there is a negative correlation. The y-axis has different scale range to well highlight the TWS dynamics in the different areas.

## 5. Conclusions

This study analyzed snow changes in the HMA region by using MODIS products from 2000 to 2018 and compared them with the trends at global level. Moreover, water resources were addressed by using GRACE TWS data at global level and in the HMA region. The main findings for the snow

changes detected in this study are in good agreement with other studies for the global level and for HMA regions [13,15,77], as well as for the TWS changes with GRACE [40,81]. On the other side, this study addressed for the first time the quantification of the snow contribution to TWS in mountain areas at global level.

The results indicate that:

- At a global level, considering significant changes, 78% of the areas show a snow decline and this percentage raises to 86% in the HMA region.
- At medium elevation, positive and negative changes in different snow parameters can be found, while at elevations higher than 4000 m a.s.l. only negative changes are detected.
- Around 50% of the areas in the HMA region and 30% at global level are suffering from significant TWS decrease. In HMA region, this decrease involves around 54% of the areas during MAM period, while at a global level the percentage of areas stays between 25% and 30% for all the seasons.
- TWS positive trends are found for maximum 10% of the areas in HMA region and for more than 20% of the areas at global level.
- In HMA region, significant changes in TWS are found especially in the southern part, involving mountain areas such as Shaluli Shan and Daxue Shan, which are also strongly affected by snow decline.
- Overall, a significant contribution of the snow mass changes to the TWS dynamics up to 30% of the areas was found during winter and spring period over 2002–2015.

This analysis revealed a high variability of snow at a global level and in the HMA region, even though the negative trends prevail. The consistent relationship between SCA and TWS changes in winter and spring period indicates the importance of snow accumulation and melting processes, thus increasing the potential impact of future climate changes, especially in dry and at high elevation areas. The strong relationship between SCA and TWS changes in areas such as the HMA, South America or Zagros Mountain needs to be monitored as these regions support large population. Moreover, in some cases, as for the Zagros Mountain, the main source of water derives from snow melting [10]. As indicated in Immerzel et al. (2020) [37], mountain ranges like Karakorum, Hindu-Kush, Ladakh and Himalaya, whose basin refers to the Upper Indus basin, are considered one of the most critical water tower units, because of threat on water availability due to a combination of a densely populated area and intensively irrigated downstream plains. The implications of the hydrological regime changes can be manifold and with cascading effects. The changes may impact natural hazard risks such as flood and drought, ecosystem functions in relation to vegetation and wildlife dynamics, and human activities as the production of hydropower energy and the irrigation for crop growth.

In this view, the results and the information provided in this study can be further updated as soon as other data become available and can be a starting point to address and quantify the impacts of snow and water resource changes in these sectors, from agriculture to energy production, to leisure activities. Moreover, the availability of water needs to be understood in its implication for the mountain environment and its biodiversity.

**Funding:** This research received no external funding.

**Conflicts of Interest:** The author declares no conflict of interest.

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
