# Peer review of "Observing Snow Cover and Water Resource Changes in the High Mountain Asia Region in Comparison with Global Mountain Trends over 2000–2018"

_remotesensing, doi:10.3390/rs12233913_

Round 1
Reviewer 1 Report
dear Editor(s),
the manuscript is clearly written, it has good structure and reads well.
I believe, there is no need to repeat what the objective of the manuscript is and what are the results found, because all this is clearly presented. The only critical remark is that potentially, the manuscript does not fit to the scope of the remote sensing journal, as it does not clearly relate to some advance in remote sensing (either methodological or application).I have only two comments:
1) The first part of the results is already presented in previous work of the author. The new part is mainly the comparison and correlation of snow parameters with GRACE observations. So I think this needs to be clearly indicated in the formulation and demonstration of the novel contribution.
2) I'm not sure whether the manuscript, in its current form, is within the scope of the journal. There is no methodological advance in application of Remote Sensing techniques and the novel contribution is more related to the link of snow characteristics to water resources. So perhaps an audience of journal (such as Water) will be more appropriate for presenting the results.
Specific comments
1) There are several " Error! Reference source not found" in the text.
2) Figure caption/label "number of areas (%)" does not read well. Do you mean Relative frequency?
Author Response
I thank the reviewer for the comments to the manuscript. Below the detailed answers to the comments are provided.
- The first part of the results is already presented in previous work of the author. The new part is mainly the comparison and correlation of snow parameters with GRACE observations. So I think this needs to be clearly indicated in the formulation and demonstration of the novel contribution.
Answer: thanks for the comments! A new sentence has been added at the end of the introduction to further underline the novelty of this paper which is related to (line 146-148):
- highlight and details on the HMA areas which were not introduced in the paper Notarnicola 2020 as this paper deals with global mountain areas. HMA is a very interesting area as it is the water tower to densely populated areas.
- correlation with water resources: this novel part has the aim to quantify how the changes in snow can impact the water resources.
2) I'm not sure whether the manuscript, in its current form, is within the scope of the journal. There is no methodological advance in application of Remote Sensing techniques and the novel contribution is more related to the link of snow characteristics to water resources. So perhaps an audience of journal (such as Water) will be more appropriate for presenting the results.
Answer: I understand the point. The paper presents results not related to the methodology rather to the exploitation of different time series of remotely sensed data to highlight the process happening in global mountain areas. I still believe that this is within the scope of the Remote Sensing journal as it shows the advancement that can be made by fully exploitation of remotely sensed data. The advancements can be related to the methodology and as well to how the data are using to get new insights. Among the scope of the journal, Remote Sensing applications are considered, as mentioned in the journal website “…to its application in geosciences, environmental sciences, ecology and civil engineering.”
Specific comments
- There are several " Error! Reference source not found" in the text.
Answer: they have been fixed
- Figure caption/label "number of areas (%)" does not read well. Do you mean Relative frequency?
Answer: yes. it is intended the number of areas affected with respect to the total number of areas. the total number of areas can be found in section 2.1. To clarify this, the caption has been changed.

Reviewer 2 Report
This is a very interesting manuscript presenting new and relevant information, data and findings. The topic of the manuscript fits well into the scientific scope of the journal Remote Sensing.
The work is well presented and carefully discussed.
I have only a few requests for minor modifications:
- Please consider to be a bit more specific on possible wider implications of your very interesting findings. Maybe a few sentences could be added here (in the discussion and conclusions sections).
- There are some errors with referencing in the manuscript text ("Reference source not found"). Please correct this accordingly.
- Please replace "et al." by the names of all co-authors at different places in the list of references.
Author Response
I thank the reviewer for the comments to the manuscript. Below the detailed answers to the comments are provided.
Please consider to be a bit more specific on possible wider implications of your very interesting findings. Maybe a few sentences could be added here (in the discussion and conclusions sections).
Answer: a few sentences have been added in the discussion and conclusions (line 618-622, line 679-683)
There are some errors with referencing in the manuscript text ("Reference source not found"). Please correct this accordingly.
Answer: They have been fixed
Please replace "et al." by the names of all co-authors at different places in the list of references.
Answer: They have been added
